

# Brief communication: Evidence of the impacts of climate extremes on power system outages in India

Jasper Verschuur[1], Srijith Balakrishnan[1]

[1]Department of Engineering Systems and Services, Faculty of Technology, Policy and Management, Delft University of
Technology, Delft, 2628BX, the Netherlands

*Correspondence to*: Jasper Verschuur (J.Verschuur@tudelft.nl)

**Abstract.** Electricity systems are prone to climate extremes such as heatwaves and cold spells, precipitation, high winds, and
flood inundation. Yet, the impacts of these climate extremes on the provision of electricity is scarce, in particular in the Global
South context. Here, we combine four years of daily electricity outages data from 370 locations across India with temperature,
wind, precipitation and flood inundation data to provide empirical evidence of their impacts of the electricity system. We find
that outages minutes can increase 20-70% during days with high wind speed (>50 m/s), 80-220% during days of intense
precipitation (>40 mm/day), and around 15-60% during heatwaves (>40 degrees Celsius). In terms of flooding, we find that
severe flood inundation in urban areas can increase daily outage minutes with a factor 2.1 to 5.5. Our findings highlight how
high frequency data can help empirically validate how climate extremes can affect essential services to customers, and how
these impacts differ across types of locations. This information is key for those countries that aim to meet universal access to
energy in the coming decades, yet at the same time will experience more frequent and intense climate extremes.



## 1 Introduction

Over 700 million people globally still lack access to electricity, most of which concentrated in Sub-Saharan Africa and South
Asia (IEA, IRENA, UNSD, World Bank, WHO, 2024). Even for those connected to the grid, power system reliability is often
low in Global South countries. For instance, grid sensors measuring voltage levels in Tanzania and India found average outages
of 400 to 1200 hours per year, equivalent to 1 to 3 hours each day (Ferrall et al., 2022). A recent study found that 1.2 billion
people are considered energy poor, meaning that they lack access altogether or do have access but without evidence of
electricity usage (Min et al., 2024).


Extreme weather-event and natural hazards can further exacerbate power system reliability. High temperatures, for instance,
can lead to power outages given the combination of transmission losses, equipment failures, and higher power demand (Do et
al., 2023; Santágata et al., 2017). Similarly, flooding can disrupt electricity services due to damages to electricity infrastructure
such as power stations, substations and transformers (Verschuur et al., 2024; Ye et al., 2024). In the decades to come, climate
change will likely increase the frequency of extreme weather events that can increase damages to electricity assets, leading to
more frequent asset failures. These localised failures can propagate through the electricity system and cause cascading impacts
to initially unaffected parts of the system, together resulting in outages to households and firms (Hallegatte et al., 2019).
Frequent electricity outages have shown to have wider social and economic impacts to its users, in particular in low and middle
income countries, such as in terms business output (Moyo, 2013), household income (Chakravorty et al., 2014) and health
(Adair-Rohani et al., 2013; Mango et al., 2021).

Despite the recognition of the vulnerability of power systems to extreme weather and natural hazards (hereafter climate
extremes), empirical evidence of the impacts of these extremes on electricity outages is scarce, particularly in the Global South.
This is mainly driven by the lack of high frequency, geolocated, data on power outages that can be matched with the occurrence
of extreme events. In this paper, we use over 227,000 daily records of electricity outages, covering 370 stations across all of
India for four years (2015-2018), and combine this with daily data on temperature, wind speed, precipitation and flood
inundation for the same time period. This allows us to infer the impacts of climate extremes on daily electricity outages.

Our results highlight a noticeable impact of all climate extremes on electricity outages. However, it also emphasises that
different types of infrastructure systems (e.g., urban versus rural) respond differently to climate extremes, and that these
climatic impacts occur on top of differences in baseline service reliability. This paper calls for more data collection on
infrastructure service reliability at the user level, and stresses that efforts the coming decades should be targeted to both achieve
more reliable services, as well as making these infrastructure systems climate-proof.



## 2 Methods

### 2.1 Electricity outage data

We consider observed electricity service data from the Electricity Supply Monitoring Initiative (ESMI), an effort by the Prayas Energy Group, which consists of minute-level voltage data for around 500 monitoring locations across India. We select all data for the period 2015 to 2018 (see Section 2.2) and only select locations with at least six months of observations. We aggregate the minute data to daily outage minutes (i.e., the number of minutes per day when a voltage below 130V was measured). This leaves us with a dataset of 227,157 of observation days across 370 observation locations (see Fig. 1a-b). More details on the ESMI dataset can be found in previous work (Canares et al., 2017). These monitoring locations are split into four categories (as provided by ESMI): (i) State Capital (urban), (ii) District Headquarters (urban), (iii) Other Municipal Area (peri-urban), and (iv) Gram Panchayat (rural). Given that the outage characteristics vastly differ per category, we split the dataset into four subsets for the analysis.

It is important to note that the exact locations of the monitoring stations is not provided, only the state and district they are located in. Hence, we have geocoded all locations manually using their location names available in the dataset. Each location name in the ESMI dataset includes the name of the village, town or city, which are geographically much smaller than the districts, and we used for the geocoding. However, the geocoded locations are approximate (few hundred meters to a kilometre) and hence it should be kept in mind that this could introduce spatial bias.

### 2.2 Temperature, precipitation, wind and flood data

We consider the impacts of four types of climate extremes; maximum temperature, peak wind gust, total precipitation and flood inundation. Daily gridded data on total precipitation is extracted from the Indian Meteorological Department gridded datasets (at 0.25 degrees at the equator) (Pai et al., 2014). Daily maximum temperature and peak wind gust are extracted from ERA5 (0.1 degrees at the equator) (Hersbach et al., 2020) via the Open-Meteo Weather API (Zippenfenig, 2023). The weather data is merged to the monitoring stations based on the closest distance to each grid centroid.

Moreover, we collected inundation maps of past flood events between 2015 and 2018 from the Global Flood Database, a global dataset consisting of mapped footprints and flood duration (per 250 meter grid cell) based on MODIS data between 2000 and 2018 (Tellman et al., 2021). There are a total of 27 flood events within the time period covered, of which 15 covering locations for which we have outage information (see section 2.3). Per flood event, we estimate the inundation per flood day (i.e., day after the start of the flood event as defined in the Global Flood Database) based on the grid-cell flood duration (e.g., if a grid cell has a flood duration of three days, we assume it was flooded on day 1, 2 and 3). We do this for four buffer areas around our best estimate of the geolocation of the monitoring stations; 2.5km, 5km, 10km, and 20km. This provides us with a time series per station on the flooded area. We use the different buffer areas to capture how both local and more regional flooding





can result in outages, given that flooding can damage regional electricity infrastructure which can result in cascading failures to the electricity distribution and hence outages at the monitoring stations.

## 2.3 Analytical approach

We use fixed-effects (FE) panel regressions, as is common practise in the climate econometrics literature (Hsiang, 2016; Kotz et al., 2021), as our main method to identify plausible causal effects of climate extremes on daily minutes of electricity outages. Daily outages minutes are represented as counts, with the distribution being over-dispersed with a rightward skew (i.e., a lot of observations with low daily outages minutes and only a small number of observations with high daily outage minutes). This makes ordinary least square regression inadequate. Instead, we use a negative binomial regression for our estimation, which is able to deal with over-dispersed count data. Although negative binomial regression is not well suited for count data with an upper bound (which we have in our data, namely 1440 minutes), the heavy skew of the data means that the upper bound is rarely reached (the 95th quantile is 733 minutes). With this limitation in mind, we still use this type of regression. We set up separate FE models for the four categories of monitoring stations.

To evaluate the impacts of maximum temperature on outage, we follow the approach proposed in previous research (Deschênes and Greenstone, 2011), which is usually used to measure the impact of temperature on electricity consumption (Harish et al., 2020; Liu et al., 2021). In line with other literature, we measure the impact for several temperature bins to capture the non-linear effects, in this case temperature bins of 4 degrees. As such, the impact of maximum temperature ($T$) on daily ($d$) outages minutes ($O$) at each monitoring station ($s$) is measured as:

$$O_{d,s} = \sum_b \beta_b T_{d,s}^b + \alpha_1 P_{d,s} + \alpha_2 W_{d,s} + \sigma_d + \delta_{sy} + \epsilon_{d,s} \tag{1}$$

With $P$ the daily total precipitation, $W$ the daily mean wind speed, $\sigma$ the day-of-the-week fixed effect (to account for varying electricity consumption), $\delta$ station-year fixed effect to account for local (time-invariant) differences in outages and trends per station. The temperature is divided into seven bins ($b$) (<20°C, 20-24°C …, 40-44°C), with the lower and upper bounds chosen to ensure sufficient data in these bins. The reference bins is set to 24-28 degrees Celsius, which for India is the most frequently occurring temperature band.

To measure the effect of wind speed and precipitation on daily outages, we use a similar approach:

$$O_{d,s} = \sum_b \beta_b P_{d,s}^b + \alpha_1 T_{d,s} + \alpha_2 W_{d,s} + \sigma_d + \delta_{sy} + \epsilon_{d,s} \tag{2}$$



For precipitation, we created five precipitation bins (0-20, 20-40, 40-60, 60-80, >80 mm per day) and the beta coefficients are evaluated with respect to the days without precipitation. In a similar fashion, the impacts of wind speed on outages can be written as:


$$O_{d,s} = \sum_b \beta_b W_{d,s}^b + \alpha_1 T_{d,s} + \alpha_2 P_{d,s} + \sigma_d + \delta_{sy} + \epsilon_{d,s} \tag{3}$$

but with five bins (20-30, 30-40, 40-50, 50-60, >60 m/s) with respect to the baseline of limited wind (peak gust of 0-20 m/s).

We next estimate the effect of flooding ($F$) on electricity outages, controlling for daily temperature, wind speed, precipitation and day-of-the-week and station-year fixed-effects:

$$O_{d,s} = \sum_b \beta_b F_{d,s}^b + \sum_c \beta_c P_{d,s} F_{d,s}^c + + \alpha_1 T_{d,s} + \alpha_2 P_{d,s} + \alpha_3 W_{d,s} + \sigma_d + \delta_{sy} + \epsilon_{d,s} \tag{4}$$

where $F$ is the flood dummy per buffer size (1 if flooding is observed). We consider two bins of flooding as a fraction of the total area per radius (0-20%, >20%). The interaction term between precipitation and flooding takes into consideration that flood inundation can in some instances happen because of local extreme precipitation while in other instances because of river flooding without local precipitation. The latter regression is only performed for the State Capital category given that the number of flood observations is too low for the other categories to meaningfully estimate the effect of flooding on outage minutes
(1.2% of data for State Capital stations has flooding observed for a 10 km radius compared to 0.07-0.15% for the other categories).





# 3 Results

## 3.1 Descriptive statistics electricity outages

Electricity outage minutes differ considerably across the four categories considered (State Capital, SC, District Headquarters, DH, Other Municipal Area, OMA, Gram Panchayat, GP). Per monitoring station in our data, we calculate the average daily outage minutes, the number of days without outage, and the standard deviation of daily outages minutes (Figure 1c-e).

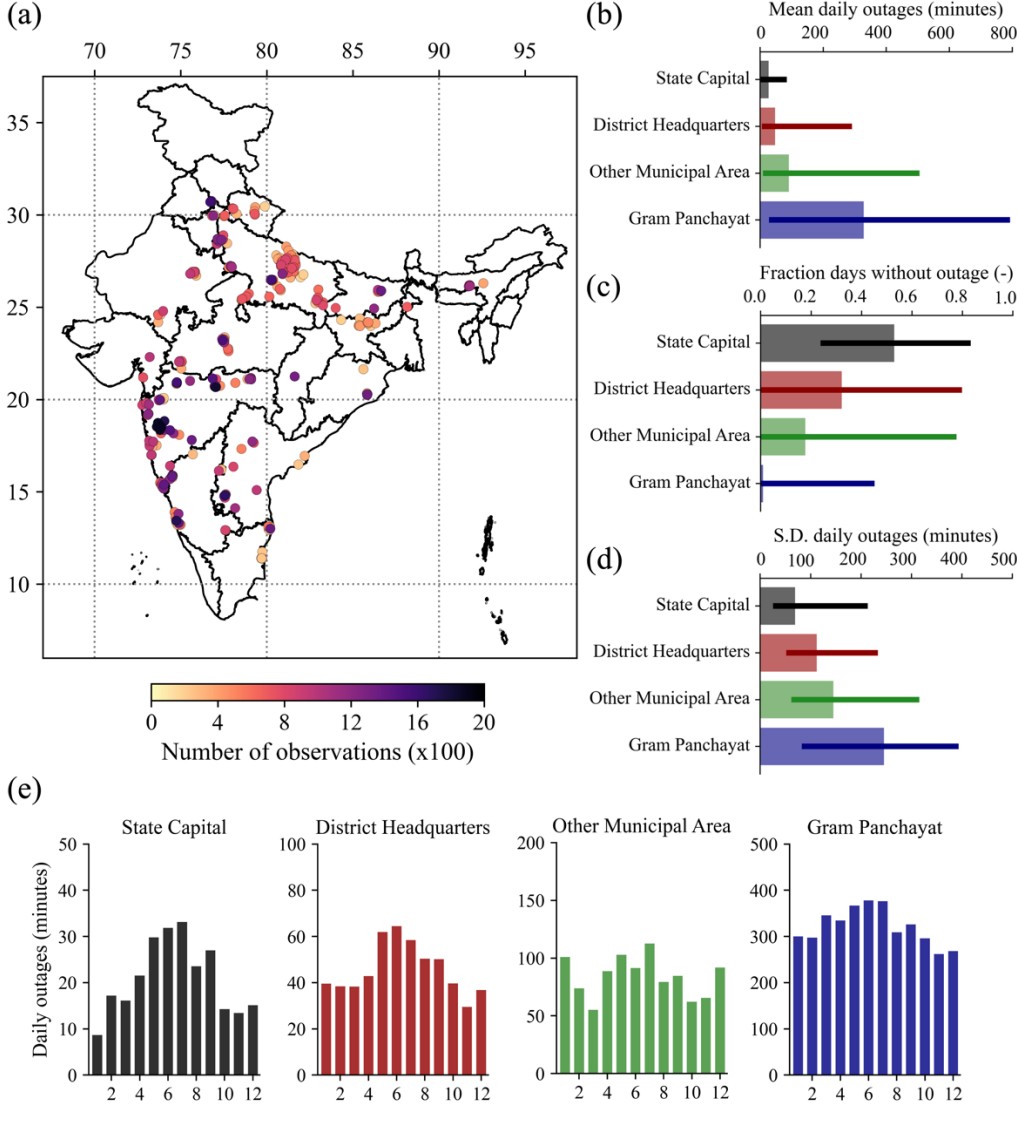

Figure 1: (a) Number of observations per monitoring over the period 2015-2018. (b) Mean daily outages across monitoring stations. The bar indicates the median value and the error bands indicate the 5[th] to 95[th] percentile across the stations. (c) Same as (b) but for the fraction of days without outage. (d) Same as (b) but for the standard deviation of daily outages. (e) Monthly daily outage of the median monitoring station.



The median (5-95th percentile), across monitoring stations, average daily outages are 27.4 (5.7 – 78.4), 47.2 (14.6 – 282.2),
103.8 (18.0 – 496.2) and 328.3 (36.7 – 783.8) minutes for monitoring stations characterised as SC, DH, OMA and GP,
respectively. Similarly, the percentage of days with zero minutes of electricity outages is equal to 53.0% (24.0 – 85.6%), 32.1%
(1.1 – 78.7%), 16.6% (0.0% - 75.7%) and 0.9% (0.0% - 44.0%) for SC, DH, OMA and GP, respectively. In other words, while
urban consumers rarely experience outages, rural consumers face intermittent supply on a daily basis. In a similar fashion, the
day-to-day variability of electricity outages, measured in terms of the standard deviation over the measurement period per
monitoring station, differs per category. SC monitoring stations have a median (5-95th percentile) variability of 75.3 (31.9 –
207.5) minutes per day, while DH monitoring stations 112.4 (57.3 – 227.7) minutes per day. For OMA and GP monitoring
stations, this is equal to a median (5-95th percentile) variability of 146.3 (67.5 – 309.4) minutes per day and 245.3 (88.2 –
388.0) minutes per day, respectively.

A clear seasonality is visible for the SC stations (Figure 1e), which follows the seasonal temperature profile across India with
high temperature during May-July and low temperature during November-February. The strong seasonality in outage is linked
to the electricity consumption patterns, which strongly follow temperature in urban areas, mainly reflecting cooling demand
during high temperature days. This same seasonality is visible for DH stations, with peaks during May-July. For the OMS and
GP stations, there is some seasonality, though less pronounced (in relative terms), likely because of lower energy consumption
variability across the year. For instance, the penetration of heating and cooling devices is much lower in rural India compared
to urban systems (World Bank, 2022).

### 3.2 Influence of wind, precipitation and temperature on electricity outages

Using Eq. (1-3), the influence of wind, precipitation and temperature on electricity outages is derived, as shown in Figure 2.
Given that we use negative binomial regression models, the beta coefficients are expressed as percentage changes; 100% means
that the count of daily outage minutes are double as high compared to the baseline. This should thus be interpreted against the
backdrop of differences in baseline outage minutes (Fig. 1b).

For wind (Figure 2a), the different system categories are relatively robust again lower wind gusts (<30m/s), in particular the
SC and GP categories. However, for all systems, we observe a positive impact of wind gusts above 50 m/s on outage minutes,
which ranges from around 25 – 70%, depending on the category. Under very extreme conditions (>60 m/s), the largest impact
is observed for the DH category, and lowest for the GP category.

Precipitation has a strong impact on daily outage minutes, although with differences observed across different categories of
power systems (Figure 2b). Low precipitation intensity already has a meaningful impact on outage minutes, which increases
steeply for higher precipitation intensities. Days with 0 to 20mm of rainfall will have 30-40% higher outage minutes, which



increase to a factor 80-200% for precipitation of 40-60 mm per day and more. For the SC category, the curve decreases after reaching 60mm per day, although this is likely associated with relatively low observations for these high thresholds. For DH, GP and OMA categories, the impact of precipitation increases with severity. The largest relative impact is observed for DH systems, with a precipitation of 80 mm per day or higher associated with over three times higher outage minutes, against the

backdrop of lower baseline outage minutes (compared to the other two categories). The curve for GP stations shows a linear relationship, with very heavy precipitation (80 mm per day) associated with close to 2.7 times higher outages compared to days without precipitation. An increasing number of GP systems are powered by photovoltaic systems, and increasingly so in more recent years (Conevska and Urpelainen, 2020). The sensitivity of outage minutes with respect to precipitation for GP could be associated with the presence of photovoltaic power systems, which generally have low power output during days with

high cloud coverage, among other factors (e.g., lower engineering standards, lack of maintenance, further away from substations). Given that climate change is expected to increase the number of days with high precipitation across India (Mukherjee et al., 2018), this will have a knock-on effect on the reliability of electricity services.

In terms of temperature, we observe again similarities across the different categories (Fig. 2c). For the SC category, the

relationship between temperature and outages increases steeply above 24 degrees (Fig. 2a). This upward linear relationship is likely driven by the cooling demand in cities and efficiency losses, both having a positive relationship with increasing temperatures. For the three other systems, we observe a U-shaped curve, similar as seen in curves relating temperature to electricity consumption in India (Harish et al., 2020). Outages can increase by 20% to 50% during heat waves (>40 degrees) because of high demand, transmission losses, and heat-related infrastructure failure. Moreover, outages increase with 10-20%

during colder days, associated with residential heating demand. The lowest sensitivity is found for the GP and OMA categories, which may be related to the variability of solar radiance in India, which varies strongly with temperature, but also lower temperature-related energy consumption changes as shown in Figure 1e.




**Figure 2: (a)** The percentage change in daily outage minutes (OM) for different peak wind gust bins, all with respect to the outage minutes observed under a 0-20 m/s day. **(b)** The percentage change in daily OM for different precipitation bins, all with respect to the outage minutes observed without precipitation. **(c)** The percentage change in daily OM for different maximum temperature bins, all with respect to the outage minutes observed for a 24-28 degrees Celsius day. The bold line indicate the mean value whereas the shaded area covers the 5th to 95th percentile of the estimated coefficients.



### 3.3 Impact of flooding on electricity outages

The 15 flood events in our combined flood and outage database results in a total number of 145-467 flood observation days, depending on the radius (2.5 – 20km), across the SC data subset (N = 33,937). The results of regressions Eq.4 are included in Table 1 per radius. Again, coefficients should be interpreted as percentage changes, e.g., a beta of 0.2 implies that any day with flooding increase the daily outage count with a factor 1.22 ($e^{0.2}$). As mentioned before, we distinguish between minor to moderate flood inundation (0-20% of area) and severe flood inundation (>20%).


In line with the results discussed in Section 3.2, we find a positive and statistically significant influence of daily weather conditions (maximum temperature, wind gust and precipitation) on outage minutes. Moreover, we find a positive and statistically significant effect of severe flood inundation on daily outage minutes for all radius sizes. The beta coefficients range between 0.7 and 1.7, meaning that any day with severe flooding has 2.1 – 5.5 times higher outages minutes compared

to days without flooding. The largest effect is for the 2.5km radius. In other words, the effect of flooding is stronger for more localised flooding compared to regional flooding, as expected. Still, severe regional flood events can increase the outage minutes disproportionately within urban power systems, likely the result of more pronounced cascading failures that may occur during these widespread flood events. We also find a statistically significant interacting term for severe flooding and precipitation for all four radius cases considered. In other words, the higher the precipitation, the lower the impact of flood

inundation on power outages, given the compounding impact of precipitation and inundation on power outages. This is to be expected given that days with high flood severity are also more likely to be associated with higher local precipitation. It should be noted that there is a certain level of error associated with the location coordinates of outage observations, which may range from a few hundred meters to several kilometres. This uncertainty should be considered when interpreting the results.

**Table 1: Results of the regression formulation (Eq. 4, Model 1-4). *P<0.1, **P<0.05, ***P<0.01. Number of observation is 33,937 with N flood the number of observations with flood inundation observed.**

| Model | (1) | (2) | (3) | (4) |
|---|---|---|---|---|
| **Radius (km)** | **2.5** | **5** | **10** | **20** |
| Max. temperature (°C) | 0.032*** | 0.032*** | 0.032*** | 0.032*** |
| Precipitation (mm/day) | 0.017*** | 0.017*** | 0.017*** | 0.017*** |
| Wind gust (m/s) | 0.008*** | 0.008*** | 0.008*** | 0.008*** |
| Perc. flooded [0:20%] | 0.222* | 0.242*** | 0.185*** | 0.112* |
| Perc. flooded [>20%] | 1.737*** | 0.686* | 1.044** | 1.325** |
| Flood perc. [0:20%] x Prec. | -0.038*** | -0.000 | -0.001 | 0.005 |
| Flood perc. [>20%] x Prec. | -0.421*** | -0.164*** | -0.544*** | -0.554*** |
| N flood (#) | 145 | 303 | 420 | 467 |
| Pseudo $R^2$ | 0.552 | 0.552 | 0.552 | 0.552 |



## 4 Moving forwards

In this research, we provided empirical evidence of the impacts of climate extremes on power outages across India. Based on the findings of our case study, we highlight three aspects to that can help improve climate risk modelling of critical
infrastructure networks and the development of climate resilience strategies.

First, it becomes apparent from the impacts of wind, temperature, and precipitation that different types of electricity systems vary in their vulnerability to climate extremes. Across the hazards, the impact of extreme precipitation is found to most impactful for outages minutes across all four categories. Still differences are also observed. For instance, in relative terms,
rural power systems are found to be more resilient against high or low temperatures and wind compared to urban power systems. The impact of heatwaves on outages is largest, in relative terms, for urban power systems. Yet, in most climate impact modelling, the vulnerability curves that relate the hazard intensity to impact on the infrastructure assets does not vary across types of systems, given a lack of data (Nirandjan et al., 2024). More empirical evidence of the impacts of climate extremes on critical infrastructure systems, as provided in our work, is needed to better understand how different types of infrastructure
systems, within countries or across countries, are able to cope with climate extremes.

Second, the reliability of infrastructure services is still low across many of the monitoring locations in our datasets, but is similar as found for other electricity systems in Global South countries (Ferrall et al., 2022), and other types of infrastructure (in particular water and telecom). More frequent climate extremes can negatively affect the reliability. In our work, we find
that systems with higher baseline reliability are more sensitive to extreme temperature, wind and precipitation in relative terms. This means that providing access to those without electricity and improving the reliability of services alone is not enough. Investments to expand and upgrade electricity systems should be complemented with efforts to make electricity systems more climate-resilient. In regions where access to reliable electricity is not the standard, solutions for a climate-resilient system may not be the most cost-efficient one (e.g., grid versus off-grid solutions). Future research should better evaluate the synergies
and trade-offs between least cost expansion pathways of energy access and improved reliability, and the resilience implications against more frequent and severe climate extremes.

Third, empirical evidence of user-level experiences of infrastructure failures are limited in practise, but provides unique insights in the differences in local-level impacts of climate hazards. In most instances, post-disaster impact assessments only
cover the total number of infrastructure damages, such as length of roads that need to be rebuild or the number of people without electricity or water. However, how these impacts differ regionally and per electricity system, including the recovery pathways after the shock, is usually unknown. Yet, previous research has highlighted how large disparities have been observed across space, for instance after Hurricane Maria disrupted Puerto Rico's power system (Román et al., 2019). High frequency data on infrastructure service provision, such as collected under the ESMI initiative, are invaluable data sources to improve



our understanding of the user experiences of infrastructure services disruptions, and how they differ across regions and communities within regions.

Our three recommendations call for a better monitoring of infrastructure services in the Global South context to increase our understanding how climate extremes, including low probability ones, can disrupt infrastructure services on top of day-to-day

reliability issues that many of these critical infrastructure systems already face. With billions of dollars needed to expand infrastructure services to meet the Sustainable Development Goals and adapt to climate change (McCollum et al., 2018; Rozenberg and Fay, 2019), it is imperative that infrastructure solutions are designed to address both policy objectives simultaneously.

**Data availability**

The ESMI station data is available at https://dataverse.harvard.edu/dataverse/esmi. The Indian precipitation data can be downloaded from the India Meteorological Department https://www.imdpune.gov.in/cmpg/Griddata/Rainfall_25_Bin.html. The wind speed and temperature data is extracted from Open-Meteo platform (https://open-meteo.com/), while flood data is taken from the Global Flood Database (https://global-flood-database.cloudtostreet.ai/).

**Competing interests**

The authors declare no competing interests.

**Author contribution**

JV and SB conceptualized the analysis. JV led the formal analysis with input from SB. JV wrote the first draft. SB provided feedback and edits to the manuscript.

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
