# Peer review of "Brief communication: Evidence of the impacts of climate extremes on power system outages in India"

_EGUsphere, 2024_

## Author Comment (AC3)

**REVIEWER 1**

This article examines the impacts of weather extremes on power outages in India. The analysis uses data of 227,157 observation days for the period from 2015 to 2018, representing 370 locations across the country. The extremes that have been considered include maximum temperature, peak wind gust, total precipitation and flood inundation.

The correlation established in this study could serve as a crucial input for policies aimed at strengthening the resilience of power infrastructure.

However, I believe including northern stations would have provided valuable insights into an important aspect of extreme weather i.e., the form of precipitation (e.g., snowfall), which causes huge disruption to power supply and damage to infrastructure during the winter months in some northern parts of the country.

Moreover, placing greater emphasis on seasonal patterns of extreme weather and their impacts on system components such as power stations, transmission lines, and transformers could provide a more detailed understanding of the relationship between extreme weather and power outages in India.

Overall, I consider the article as a noteworthy contribution.

*We would like to thank the reviewer for their insightful comments and feedback. We overall agree with the points raised.*

*Indeed, snowfall is a major climate-related event that can cause electricity outages in northern India. We have included precipitation as one of the variables in our models, which include both rainfall and snowfall. Unfortunately, the station data in the northern provinces of India has a sparce coverage, both in spatial and temporal dimensions. This prevented us from separating snowfall and rainfall. Similarly, it is preventing us from accurately looking at the impact of low temperatures on outages. We agree that this could be studies more.*

*We have added this now to the text:*
*"L79: Other weather variables could also be important for power outages, such as snowfall in the northern States of India. However, the power outages observation data is too scarce in the northern States to accurately infer the effect of snowfall on power outages."*

*In a similar fashion, while we look at the relationship between extreme weather and power outages, we cannot show what is causing these outages, i.e., whether failures to power station, transmission lines of transformers. Bringing in additional data to better comprehend the failure pathways should be important future research.*

*We have incorporated the second point in the text as future research direction:*
*"L252: In particular, a better understanding of the failure pathways of different types of systems, whether due to failure of power supply, transmission lines, substations or transformers, is needed to improve climate impact modelling."*

**REVIEWER 2**

This manuscript examines the impact of climate extremes (heatwaves, high winds, heavy precipitation, and flooding) on electricity outages across India. Using four years (2015–2018) of high-frequency outage data from 370 locations, combined with climate data, the study employs fixed-effects panel regressions to quantify outage increases during extreme weather events. Findings indicate that outages rise significantly—by up to 220% during heavy rainfall and up to 5.5 times during severe floods. The study highlights vulnerabilities in power infrastructure, particularly in urban areas, and calls for improved data collection and climate-resilient energy investments in the Global South.

Although such a study is required for climate resilience planning, the current version has significant limitations. Fix the grammatical and sentence errors throughout the work. Write the whole manuscript more concisely. Below are some more specific comments for further improvement.

*We would like to thank the reviewer for their insightful comments and feedback. We overall agree with the points raised and have tried to address them to the best of our ability, and within the constraints of the Article Type. We have also done a careful grammatical/sentence check to improve readability. See our detailed response below.*

1. In the abstract, this line '….80-220% during days of intense precipitation….' is written (similar to the 3.1 section '…..increase to a factor 80-200% for precipitation of 40-60 mm per day…'), how did the author calculate and find 220%?

*Many thanks for pointing out this inconsistency. Indeed, they refer to the same numbers and hence should be the same in the text. This range is derived from the range across the estimated models in Figure 2.*

*We have changed the text to 80-220% consistently.*

2. There is redundancy in the introduction (paragraph 3) and methods (2.1 section) section. What are the current research gap(s) and novel contribution(s) of this work?

*We have briefly added the current research gap and contribution at the end of the introduction.*

3. In the abstract and introduction, the author mentioned about 370 power point stations. However, in section 2.1 the author mentioned both 370 and 500 stations. Observation days are mentioned in the introduction 227000; but in the 2.1 section 227157.

*Many thanks for pointing out this consistency. We have now made the number consistent throughout. We end up using 370 observation stations, with together have 226,974 observation days, after filtering out the 370 stations from the 500 in total.*

4. The paper highlights the scarcity of empirical evidence, but does it truly provide novel insights for the global south, or does it mainly confirm what is already expected based on theory and prior studies?

*Indeed, as the reviewer rightfully points out, our study confirms what is likely to be expected based on theory and prior studies. Yet, empirically confirming what is theoretically expected is still valid research in our view. As now more clearly written in the research gap, an analysis like we do has not yet been done before in the Global South context given a lack of daily outages data across large spatial scales. Our study is the first that does so, despite the constraints of the data and analysis identified.*

5. How likely is the accuracy of the ERA5 data compared to the station data?

*The accuracy of ERA5 data compared to station data depends on the region. First of all, ERA5 is calibrated on station data. In fact it is a data assimilation product of station data and satellite observations. For India, one study (Mahto and Mishra, 2019) showed that overall ERA5 outperform any other data product, agreement is good, although with some small biases. We have added this reference as a justification of using ERA5 in the paper:*

*"L76: Previous research has shown that ERA5 is the best performing gridded reanalysis product across India, showing good agreement between modelled estimates and observations (Mahto and Mishra, 2019)."*

6. Given the interannual variability of extreme weather, a longer dataset (e.g., spanning multiple decades rather than only 4 years) yields more robust results.

*We agree, that would be more robust. Unfortunately, to the best of our knowledge, such datasets do not exist for the Global South.*

7. Write down the sources of data in a single table for better understanding.

*Given that we only use four main data sources (ERA5, precipitation data, outage observation data, and Global Flood Observatory data), we do not warrant it necessary to include an additional table. Moreover, the data sources are mentioned again in the data availability statement.*

8. What type of cascading failures are you talking about?

*Many thanks for pointing this out. This is indeed not well explained. We have added further clarification on the cascading failures in L28-29, L91-92 and L230-231. As an example, we refer to the impacts of say flooding of a substation that is then causing outages outside of the flooded area.*

9. Why did you select fixed-effects (FE) panel regressions for this study over other statistical methods? Is it more advantageous than other methods?

*Fixed-effect panel regressions are the standard type of models for this kind of empirical analysis in the climate econometrics literature. Similar studies that try to infer a climate-response function also use these types of models, see:*

*"L94: We use fixed effects (FE) panel regressions as our main method to identify plausible causal effects of climate extremes on daily minutes of electricity outages. FE panel regression models are common practise in the climate econometrics literature (Hsiang, 2016; Kotz et al., 2021). Compared to other estimation approaches, panel FE regressions address concerns regarding the influence of omitted time-invariant characteristics and local trends (Ortiz-Bobea, 2020). "*

10. Is the use of negative binomial regression appropriate given the upper bound of daily outage minutes (1440 minutes)?

*This is indeed an important point. This depends a lot on how the distribution looks like. If there are many observations close to 1440, this would be troublesome, as the model is likely to predict values above this range. In our cases, we have a heavily negatively skewed distribution, with only 9% of observations above 500 minutes, 2% above 1000 minutes per day, and 0.5% of observations being 1440 minutes. This still validates the use of this type of model, with the caveats mentioned.*

*In addition, we test how often our model is predicting values above this upper bound. The regression models only do so for 0.28% for Gram Panchayat, 0.02% for District Headquarter, 0.01% for State Capital and 0.05% for Other Municipal Area. Hence, this is almost negligible and should not affect the main results and conclusions.*

*We have rewritten the paragraph that explains this:*

*"L105; Although a negative binomial regression model assumes an unbounded distribution, we have an upper bound in our data (i.e., 1440 minutes). However, due to the heavy skew of the data, the upper bound is rarely reached; only 2.3% of data points are >1000 minutes and 0.5% of datapoints are full outage days. Moreover, after checking for the regression models described below (Eq. 1-4), the model only predicts 0.02-0.28% of data above our upper bound, depending on the system. It is therefore still appropriate to use a negative binomial regression model."*

11. The manuscript notes that geocoding errors could introduce spatial bias. Could this lead to misattribution of climate impacts, particularly for flood-related outages?

*The georeferencing procedure could indeed include a spatial bias, which results in the measurement error related to assigning the location of the stations. This is also why this is explicitly stated. As a result, the measurement error induced in the distance calculations could underestimate the effect of floods and result in larger standard errors leading to weaker statistical significance for the coefficients related to flood impacts.*

*However, in this study, potential geocoding errors, even if exist, will not be more than 500m in most of the cases in the dataset. Therefore, we do not expect the model results are considerably affected. But, this could be one of the reasons why we find clearer evidence of flood impacts for extreme flood inundation extends versus more localised ones.*

*For the other weather extremes, we do not believe this will impact the results as they are usually driven by synoptic weather patterns that impacts wider areas. Hence, a 0.5km spatial bias is not important.*

*We have added this further:*

*"L67: It is important to note that the exact locations of the monitoring stations is not provided, only the state and district they are located in. Hence, we geocoded all locations manually using their location names available in the dataset. Each location name in the ESMI dataset includes the name of the village, town or city, which are geographically much smaller than the districts. However, the geocoded locations are approximate (few hundred meters to a kilometre). Hence, it should be kept in mind that this could introduce a spatial bias. "*

12. The author should perform a statistical analysis (e.g., goodness of fit, sensitivity analysis) to evaluate the performance of their models.

*Many thanks for pointing this out. We agree that this should have been done. In the plots in Figure 2, we now report the Root Mean Square Error (RMSE) and R2-squared statistic for each model, as well as adding the RMSE to Table 1.*

13. While the study finds strong statistical associations, does it adequately consider non-climatic factors (e.g., grid mismanagement, underinvestment, theft) that may be equally or more important in explaining electricity unreliability?

*We account for non-climatic factors by including station-fixed effects, year fixed-effects, and day of the week fixed-effects. These are included to capture station-specific variations in things like local grid mismanagement, underinvestment and theft. Moreover, the station-year fixed effects also capture trends in these unobservable local grid characteristics.*

*In an ideal world we would have more variables to control for non-climatic factors. However, this type of data is not available, not would this be available anywhere, even in Global North countries.*

14. Why do rural power systems found more resilient against high or low temperatures and wind than urban power systems?

*These systems are not necessarily found to be more resilient, as the y-axis has to be interpreted in a relative fashion, i.e., for stations with a lower mean outages, it is "easier" to get higher relative outages minutes than for stations with higher mean outages. Hence, the conclusions that one system is more resilient than others cannot be drawn, or at least it depends whether one looks at the outage minutes in absolute or relative terms. Hence, we have tried to avoid drawing conclusions like this.*

15. The work lacks convincing justification (discussion) for the results.

*We have tried to rewrite our discussion and conclusion. However, we have to stay within the constraints of the manuscript type and journal requirements, as suggested by the editor.*

16. The conclusion section is missing.

*We have tried to rewrite our discussion and conclusion. However, we have to stay within the constraints of the manuscript type and journal requirements, as suggested by the editor.*

17. Overall, this work relies solely on fixed-effects panel regressions without addressing potential endogeneity or alternative causal inference methods. The geocoding inaccuracies and limited four-year datasets could introduce spatial and temporal biases, reducing result reliability. Additionally, the study lacks a clear distinction between direct and indirect outage causes, potentially overstating climate impacts. The negative binomial regression's limitations (upper bound issue) and potential omitted variable bias further weaken causal claims. Finally, the findings are largely confirmatory rather than novel.

*We agree that several of the concerns raised by the reviewer are valid and warrant attention. However, the primary objective of this study was to provide empirical evidence for the effect of climate extremes on consumer-level power supply reliability under different settings in the Global South, where spatial variations are often correlated with socioeconomic conditions. We were not only able to statistically highlight the spatial variations but also model the effect of climate extremes like extreme temperature, precipitation, wind and floods on consumer-level power reliability.*

*While some of the power supply reliability trends and relationships identified in the study have been previously demonstrated through exploratory modelling approaches, we do believe that the new empirical evidence can further strengthen existing theories and develop new hypotheses in the Global South contexts.*

*As far as cascading effects are concerned, an exploratory model, such as a power systems simulation model could be more suitable than an empirical model for understanding the extent of direct and cascading effects of climate extremes. However, the current study present empirical evidence which can be used for calibration and validation.*

*Therefore, we think that study is timely and important for researchers and practitioners who are interested in climate resilience of infrastructure systems in the Global South, even though the extent of analyses have been limited by the availability of data. That said, we do agree that the concerns raised by the reviewer are relevant and therefore we have acknowledged these limitations related to data and empirical modelling in the revised manuscript throughout.*